# Occupational Therapy Intervention in the Child with Leukodystrophy: Case Report

**DOI:** 10.3390/children10071257

**Published:** 2023-07-21

**Authors:** Rachele Simeon, Anna Berardi, Donatella Valente, Tiziana Volpi, Samuele Vagni, Giovanni Galeoto

**Affiliations:** 1Department of Human Neurosciences, Sapienza University of Rome, 00185 Rome, Italy; rachele.simeon@yahoo.com (R.S.); donatella.valente@uniroma1.it (D.V.); giovanni.galeoto@uniroma1.it (G.G.); 2IRCSS Neuromed, Via Atinense, 18, 86077 Pozzilli, Italy; 3Nuova SAIR Via del Tecnopolo, 83, 00131 Roma, Italy; tizianavolpi25@gmail.com; 4School of Occupational Therapy, Sapienza University of Rome, 00185 Rome, Italy; vagni.samuele23@gmail.com

**Keywords:** occupational therapy, leukodystrophies, autonomy, activities of daily living

## Abstract

Background: There are many different types of Leukodystrophies. Specifically, children with hypomyelination and congenital cataract syndrome (HCC) in addition to motor retardation development, hypotonia and progressive spastic paraplegia, associated with cerebellar ataxia and peripheral neuropathy, have early bilateral cataracts and intellectual disability as pathognomonic symptoms. HCC rehabilitation treatment is not well defined, but a significant amount of evidence in the literature has demonstrated the effectiveness of occupational therapy (OT) treatment in children with similar symptomatology. For this reason, the aim of this study was to describe the improvement in the autonomies and social participation of a child with HCC following OT treatment. Methods: A.E. was a 9-year-old child with HCC with severe intellectual disability. OT intervention lasted 3 months biweekly and each session lasted 45 min. Each session was divided into two parts: The first part aimed to increase the child’s active involvement through activities; the second part involved training in Activities of Daily living (ADL). The outcome measures were: ABILHAND-Kids; Pediatric Evaluation of Disability Inventory; Comprehensive OT Evaluation Scale; ADL and Instrumental Activities of Daily Living. Results: A.E.’s outcome measure reported an improvement from an autonomy standpoint and in the child’s general activity participation; there was also an increase in A.E.’s interpersonal skills. Conclusion: OT treatment improved A.E.’s autonomy.

## 1. Introduction

Leukodystrophies (LDs) are a heterogeneous group of disorders with widely varying clinical manifestations and pathological mechanisms, afflicting the brain’s white matter, with or without peripheral nervous system involvement [1]. These are rare diseases, affecting approximately [1] in 7500 individuals [2].

Based on the opinions of experts from the Global Leukodystrophy Initiative (GLIA) association, there are about 30 disorders that can be classified as leukodystrophy [3]. Despite the many disorders that fall under this classification, different types of leukodystrophies present similar scenarios [3]. Hypomyelinating LDs are the cluster of LDs involving the white matter. This group of disorders results in permanent damage to the myelin pattern in the Central Nervous System (CNS) [4,5,6]. In 1910, Ludwig Merzbacher, a German neuropathologist and psychiatrist, was the first to describe this sign as characteristic, following analysis of brain tissue [7].

Since MRI is not specific for diagnosing hypomyelinated leukodystrophy, and other specific, laborious, and expensive diagnostic techniques (such as exome analysis (WES)), there appears to be a major delay in diagnosis [6].

Diagnosis is even more complex in children, as CNS development is not complete in the earliest months of life. In children under 18 months, in order to verify whether the myelin damage is permanent or there is nonspecific delay in myelination, it would be necessary to perform an MRI several times every 6 months [6,7].

Also, the percentage of undiagnosed cases of LD decreased from 50% to 20–30% from 2010 to 2016 and is expected to decrease further over the years [6].

Today, by combining the results of MRI, biochemical and metabolic analysis, brain CT scans, and exome analysis (WES), we have much clearer results, although the percentage of undiagnosed cases is still high.

From a clinical point of view, leukodystrophies are often reputed to be fatal. Although this may be true for many leukodystrophies, in recent years there have been several cases of clinical stability or even permanent improvement [6].

The clinical manifestation of hypomyelination is extremely heterogeneous: comparable myelin damage on the basis of neuroradiological evidence also has more or less severe clinical manifestations, even depending on the age of the individual [8,9]. Moreover, in the majority of cases, deterioration of neurologic signs and symptoms is slow and is followed by a long period of clinical stability. In patients with early onset, deterioration usually begins in late childhood or adolescence [10].

In most children with LD, the clinical manifestation is predominantly motor, characterized by delayed motor development or regression. The frequency of acute exacerbation is higher in children who have concomitant pathologies with LD. Motor symptomatology is characterized by spasticity, dystonia, choreoathetosis, tremor, and cerebellar or proprioceptive ataxia. Neurological manifestations are varied and include cranial nerve involvement, autonomic dysfunction and seizures, among others. Other neurological symptoms that may be present in people with LD include macro/microcephaly, school regression, and cognitive or neurobehavioral abnormalities. Regarding extra-neurological symptoms, there may be skin involvement, dysmorphic, hair changes and involvement of the gastrointestinal system. Symptomatology involving the musculoskeletal system with orthopedic deformities may also be present [11].

The FAM126A gene in hypomyelination syndrome and congenital cataract (HCC) has autosomal mutations. The mentioned gene encodes the transmembrane protein Hyccin, which interferes in phosphoid lipid synthesis [12]. Children with hypomyelination and congenital cataract syndrome (HCC) in addition to motor retardation development, hypotonia and progressive spastic paraplegia, associated with cerebellar ataxia and peripheral neuropathy, have early bilateral cataracts and intellectual disability as pathognomonic symptoms [13,14,15].

Hypomyelinating leukodystrophy treatment is not well defined, either pharmacologically or from a rehabilitation perspective. The literature does not provide a well-defined rehabilitation protocol for this type of disorder. Numerous pieces of evidence in the literature have demonstrated the effectiveness of occupational therapy treatment in children with similar symptomatology, motor symptoms [16,17,18] and intellectual disability [19]. This case report aims to describe the improvement in the autonomies and social participation of a child with HCC following OT treatment.

## 2. Materials and Methods

This case report was conducted by a research group of Sapienza University of Rome and the Rehabilitation and Outcome Measures Assessment (ROMA) Association, who were involved in different studies on rehabilitation [20,21,22,23,24,25,26,27,28,29].

### 2.1. Patient Information

A.E. is a 9-year-old child with HCC in comorbidity with severe intellectual disability, bilateral profound sensorineural hearing loss, bilateral glaucoma and focal epilepsy. He is in the third year of primary school and has two younger brothers. He lives in an apartment and has no extracurricular activities outside of therapy. At the time of intake, the child has been performing speech therapy for two years. The moment of detachment from the reference figure (the father) occurs serenely. However, if left alone, it denotes a tendency toward isolation. He depends on others for major hygiene tasks and requires assistance in feeding, minor hygiene, dressing and undressing. He does not understand simple, contextual orders unless accompanied by gestures and/or pictures and unless previously experienced.

The patient presents difficulty recognizing medium-sized images (10 cm or so) from the medium to long distance; the difficulty is overcome by bringing the figure a few centimeters closer to the face or using high-contrast colors. His gait appears to be autonomous and functional, although he demonstrates vestibular and proprioceptive difficulties on qualitative assessment, and his gait is mildly ataxic with an enlarged base. He is only partially spatially oriented: A.E. often moves through ambulatory spaces without having a goal and does not explore his environment functionally. Functional but poorly harmonious postural transitions characterize gross-motor development. Global rigidity is present, and his motor skills in general are characterized by poor fluidity. He manifests additional difficulties in performing some movements, especially those requiring eye-hand coordination (e.g., throwing a ball). He presents generalized hypotonia in all four limbs. Communicative intentionality and motivation for the task is absent. He does not verbally and adequately communicate his needs and does not use any alternative communication system (AAC).

Due to the child’s difficulties approaching an object, difficulty in perceiving the three-dimensionality of space is hypothesized. Patterns of action on the object are poor and immature; the child tends to beat/shake/bring the object to the mouth: these are sensory patterns aimed at self-stimulation, which are repeated in a repetitive and stereotyped manner. His graphic skills are immature in relation to the child’s age and unmotivating for the child.

### 2.2. Diagnosis

A.E. has HCC and in comorbidity presents severe intellectual disability, epilepsy, and bilateral and profound sensorineural hearing loss.

Intellectual Disability (ID) is an incomplete mental development that limits a person’s general abilities compared to individuals of the same age, gender and socio-cultural context. ID is present in between 1% and 3% [30] of the global population. The *Diagnostic and Statistical Manual of Mental Disorders* (*DSM-V*) [31] groups intellectual disability under Neurodevelopmental Disorders, together with Communication Disorders, Autism Spectrum Disorder (ASD), Attention Deficit/Hyperactivity Disorder (ADHD), neurodevelopmental motor skills (including tic disorders) and Specific Learning Disorders [32]. The *DSM-V* also identifies three fundamental clinical criteria for the diagnosis: deficits in intellectual functioning, e.g., reasoning, problem solving, planning, abstract thinking, judgment, academic learning and experiential learning, confirmed by clinical evaluation and intelligence tests; deficits in adaptive functioning. These limitations occur during the developmental period.

The *DSM-V* identifies four levels of severity: mild, moderate, severe and profound [31].

The percentage of children and youth with epilepsy in the world is approximately 0.5–1% [33]. The management of epilepsy is multidisciplinary: several different health professionals are involved in diagnosis, treatment, and care. This is also due to the impact of the pathology on the quality of life of both those affected and their families [34,35,36].

About 1 in 5 children are affected by hearing loss by the age of 18. Studies show that rehabilitation plays a key role in the management of children with hearing loss, as without it, detrimental effects on these children’s language, development, education and cognitive outcomes can occur [37].

### 2.3. Therapeutic Intervention

The study was carried out at the new SAIR rehabilitation center in Rome; the occupational therapy intervention lasted 3 months biweekly and each session lasted 45 min. To ensure a quantitative baseline, all outcome measures were administered before and at the end of the intervention to record any improvements in specific domains.

Each session was divided into two parts. The first part of the session aimed to increase the child’s active involvement through activities such as drawing, painting, waxing or music. Besides increasing the child’s motivation and interest, from a therapeutic point of view they improve the motor, sensory, tactile and visual areas. They also improve eye-hand coordination. The second part involved training in activities of daily living (ADLs) and motor skills. Following a specific assessment, ADLs in which the child had reduced performance were investigated. For motor skills, motor trails and ball games were used. All strategies to improve the child’s affinity between setting and skill were implemented.

In the ADL, during the three months of therapy, a defined pathway was charted:-Hand washing: in the early stages of treatment, the child required constant supervision due to being easily distracted and his failure to use the soap dispenser correctly. In addition, he tended to want to maintain contact with the water for as little time as possible and disregarded the correct sequence for performing the task. These difficulties disappeared at the end of the three months of treatment. Episodic misuse of the dispenser, related to incorrect hand positioning that should receive the soap, remains present. In the early stages of treatment, the major problems concerned the hand drying phase, as the child could not adequately coordinate the movements of both hands. At the end of the treatment, these difficulties did not persist.-Face washing: During this activity, difficulties related to removing his glasses emerged. In fact, the child refused to remove his glasses, as without them he would have visual difficulties, preventing him from completing the activity.-Sweatshirt/jacket: the child could not remove a sweatshirt without a zipper because of the presence of the hearing aids, combined with the difficulty in coordination and movement. So, it was suggested that caregivers use only sweatshirts with zippers to make the child more independent. In the early stages of treatment, even with the zipper, difficulties related to fine motor skills and vision emerged: in fact, the child struggled to grasp the hook of the zipper and put it in the space provided in the closing phase. This difficulty was partially remedied using a hook enlarger. In this specific case, it is a lace tied to the end of the hook. The use of this aid made it easier to accomplish the task. The child has greater difficulties dressing than undressing. During dressing, even at the end of the treatment, supervision and often the assistance of an adult remained necessary.-Shoes/socks: This activity was particularly complex in the early stages of treatment, particularly during dressing. The main difficulties were related to the second stage (slipping the shoe on), particularly with the right shoe. At this stage, the child often lost balance, lifting the lower limb and unbalancing backward to impart the necessary force. So, a custom-made stool was incorporated into the activity to facilitate the task. The difference emerged from the very first moments, as in this way, A.E. was able to impart the force in one vertical direction, from top to bottom, also making use of his own weight. Residual difficulties concerning the back of the shoe, which folded in on itself, hindered the operation’s success. So, a new aid, a shoehorn, was added to prevent the scenario just described from occurring. The operator used the shoehorn, given the child’s difficulty understanding its function and, consequently, its use. In the last few sessions, it was possible to remove the shoehorn; thus, after three months of treatment, shoe dressing now occurs without difficulty and almost entirely independently. Dressing, however, turns out to be more complex even at the end of the treatment because of problems related to motor skills, coordination, and vision.

### 2.4. Outcomes

The rating scales that will be used are ABILHAND-Kids [38]; Pediatric Evaluation of Disability Inventory (PEDI) [39]; Comprehensive Occupational Therapy Evaluation Scale (COTES) [40]; Activities of Daily Living (ADL) and Instrumental Activities of Daily Living (IADL).

ABILHAND-Kids [38] is an outcome measure filled out by parents. It is made up of 21 total ITEMs that examine bimanual activities and cover many areas of daily life. For each item, parents are asked to indicate the child’s perceived difficulty on a three-level scale: Impossible (score 0), Difficult (score 1) or Easy (score 2).

The PEDI contains three scales: the Functional Skills Scale (FSS), the Caregiver Assistance Scale (CAS), and the Modifications Scale (MS). The FSS assesses deficits regarding functional skills; the CAS assesses the extent to which the caregiver is involved in care; and the MS is a frequency count of the type and extent of environmental modifications that support functional performance.

Each scale includes three domains: Self Care (SC), Mobility (M) and Social Function (SF).

The PEDI provides two different types of scores: normative standard scores (NSS) and scalar scores. The NSS covers an age range of 6 months to 7.5 years.

Scalar scores indicate the child’s performance along a continuum from easy tasks to relatively difficult tasks in a particular domain, such as self-care. In this case, the child’s age is not taken into account. The scale ranges from 0 to 100, with higher scores representing increasing levels of functional performance.

COTES is an assessment tool of 25 items divided into 3 sub-scales: I General Behavior, II Interpersonal Communication and III Task Behavior.

The 7 behaviors in the first part provide general information about the patient’s habits and routines. The 6 behaviors listed in the second part concern communication and interaction skills; these can be assessed as the occupational therapy environment allows the patient to interact with other users and operators during structured and unstructured activities. Finally, the third part of the scale comprises 13 behaviors concerning skill during the performance, a fundamental area in occupational therapy. For each of the 26 items, it is possible to give a score ranging from 0 to 4 in order of severity: 0—no problem, 1—minimal, 2—mild, 3—moderate and finally 4—severe. The total score is given by the sum of the partial score of the three sub-scales, thus obtaining a maximum total score of 104: the higher the score, the lower the patient’s skill level.

ADL involves assigning one point for each independent function to obtain a total performance score ranging from 0 (complete dependence) to 6 (independence in all functions).

For scoring, it is necessary to translate the three-point rating scale (no assistance, partial assistance, or full assistance) into the dichotomous classification “dependent/independent.” A simplified scale is also used to calculate the IADL index by assigning one point for each independent function to obtain a total performance result ranging from 0 (complete dependence) to 8 (independence in all functions).

## 3. Results

As shown in Table 1, as for the ADL and IADL scale items, the score remained unchanged.

The COTES scores were slightly improved: from 86 initial scores to 81. In item group I, “general behavior,” item B, “non-productive behavior,” changed from 4 (severe) to 3 (moderate); in item group II, “interpersonal communication,” item D, “sociability,” changed from 3 (moderate) to 2 (mild), and item E, “behavior to get attention,” changed from 3 (moderate) to 2 (mild); in item group III, “behavior during the task,” item B, “concentration,” changed from 4 (severe) to 3 (moderate); item C, “coordination,” changed from 4 (severe) to 3 (moderate). ABILHAND KIDS: in T0 in all items were rated as “impossible,” except for the item “taking coins from wallet,” which was rated as “difficult.”

In T1, several items registered improvements, in particular:

The item “wearing a hat” changed from “impossible” to “difficult,” the item “pulling up the zipper of pants” changed from “impossible” to “difficult,” the item “wearing a backpack” changed from “impossible” to “difficult,” and the item “unscrewing the cap of a bottle” changed from “impossible” to “difficult.” The item “opening the cap of toothpaste” changed from “impossible” to “difficult,” the item “unthreading a sweater” changed from “impossible” to “difficult,” and the item “turning the sleeves of a sweater” changed from “impossible” to “difficult.”

In addition, the item “get coins from wallet” changed from “difficult” to “easy” and the item “pull up the zipper of a dress” changed from “impossible” to “easy.” On the other hand, as far as PEDI is concerned, we noted an improvement in all ITEMs of the scale.

## 4. Discussion

Regarding participation, after about a month of treatment, during dressing, the patient began to use gestures to ask for his hat (which he had not yet worn) after putting on his jacket.

In small hygiene tasks, during hand washing, about one month after the start of the intervention, he started to run the water independently after about two months; after the beginning of the intervention, he started to put soap on independently, although he failed sporadically. In the drying phase, he initially rolled up the paper without drying his hands. About halfway through the treatment, through imitation, he began to dry his hands independently, rubbing the paper on the wet areas.

After about a month and a half of treatment, he said “wash” during a handwash. This event was then repeated at every single wash.

After one month, at the sight of the platform used to facilitate handwashing, he began to head independently and without prompting to the sink, demonstrating that he had learned the function of the objects used in therapy. At the end of treatment, he demonstrated that he now recognizes the outpatient environments and purposefully explores them.

After about two months of treatment, during the shoe dressing/undressing activity, he expressed his unwillingness to perform the activity through unequivocal and functional gestures, i.e., making a sound of displeasure and crossing his arms. By the end of treatment, his concentration time for the activities of his interest, which was close to zero at the beginning of the intervention, increased.

About halfway through treatment, he became able to recognize the therapeutic routine. At the end of treatment, he showed that he has correctly memorized the sequence of steps to follow during handwashing.

As for motor skill training, he quickly assimilated the various games and how they were played. He also showed much enthusiasm during these activities.

From a relational point of view, improvements are evident; in fact, about halfway through the treatment, A.E. began to perform the gesture of throwing a kiss with his hand, accompanied by an onomatopoeic sound (“muah”) when he was about to leave the room at the end of therapy. This was considered very important as a spontaneous communicative gesture.

Treatment was based on most evidence in the literature, and as reported in previous studies, we can confirm that improvements from an autonomy and social participation point of view also exist for this study [17,41,42].

It is recommended that further research be conducted with multiple patients (e.g., small n design, multiple case studies…), because the main important limit of this study is the single-subject design, but the results may be generalizable to similar cases. For this reason, it is important to encourage referral to occupational therapy services.

## 5. Patient Perspective Informed Consent

A.E.’s parents signed informed consent before the start of treatment, and the case report was conducted according to the CARE checklist.

A.E.’s parents report improvement from an autonomy standpoint and in the child’s general activity participation; they also report increased interpersonal skills.

## Figures and Tables

**Table 1 children-10-01257-t001:** Quantitative results.

	ADL	IADL	COTES	ABILHAND	PEDI-Self-Care— Practical Skills	PEDI Mobility–Practical Skills	PEDI— Social Adaptation— Practical Skills	PEDI-Self-Care— Caregiver Assistance	PEDI- Mobility—Caregiver Assistance	PEDI- Social Adaptation— Caregiver Assistance
T0	3/6	0/8	86	1	37	59	6	18	34	0
T1	3/6	0/8	81	11	47	59	10	20	34	1

T0 = before treatment; T1 = after treatment; ADL = Activities of Daily Living; IADL = Instrumental Activities of Daily Living; COTES = Comprehensive Occupational Therapy Evaluation Scale; ABILHAND = ABILHAND Kids; PEDI = Pediatric Evaluation of Disability Inventory.

## Data Availability

Data that support the findings of this study are available from the corresponding author upon reasonable request.

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
