# Peer review of "Occupational Therapy Intervention in the Child with Leukodystrophy: Case Report"

_children, 2023, doi:10.3390/children10071257_

Round 1
Reviewer 1 Report
Occupational Therapy intervention in the child with leukodystrophy: case report
This is a well-written and conducted paper that highlights the role of rehabilitation in children with Leukodystrophy. The findings show the clinical benefits of rehabilitation in improving functional performance, independence, and quality of life for the patient.
These are my detailed comments:
Abstract:
· Please use past tense
Introduction:
· Page 2, line 49: provide a reference
· Page 2, line 57/58 “it is necessary … symptoms”: the sentence seems out of context, please delete it or integrate it within the paragraph.
· Page 2, line 85: the aim of “quality of life” is not consistent with the abstract “autonomy and social participation”.
Methods:
· The patient information is well described; it is detailed and thorough.
· Page 3, line 140: the statement “hearing loss in children common” does not apply to all children. Please specify which children group.
· Please use past tense across the paper
· Page 4, lines159-192: This section contains a detailed description of the intervention, but also outcomes of the intervention (e.g., these difficulties disappeared at the end … of treatment”, ...). Although it reads well in its present form, the manuscript would be more organized if this section focused on “intervention” only. I recommend moving the outcomes to the results section.
· The use of multiple standardized outcome measures is ideal for case studies. Well done
· The procedure section should clearly state that the outcome measures were used before the intervention, and then after the intervention.
Results:
· Page 5, line 242: the statement “COTES scores appear to be slightly improved” should read “COTES scores were slightly improved”.
· Page 5, starting line 252: the paragraph is hard to follow. The T1 results would be better presented in a table.
Discussion:
· The patient is sometimes referred to as “he” and sometimes “she”. Please be consistent throughout the paper.
· The entire discussion section needs to be rewritten completely. It lacks the main component of a discussion (main findings of the study, clinical interpretation of results, future research directions, limitations). For example:
· Intervention showed significant clinical improvements
· The results may be generalizable to similar cases
· Recommendations:
o Conducting further research with multiple patients (e.g., small n design, multiple case studies...)
o Encourage referral to occupational therapy services
· Limitations:
o Single-subject design
o Possible maturation or improvements due to other factors
Author Response
Dear Editor,
We appreciate the opportunity to resubmit our article entitled “Occupational Therapy intervention in the child with leukodystrophy: case report”. We would like to thank the referees for the careful and constructive reviews. We have made corresponding changes directly to the manuscript where appropriate with changes tracked. The revised version of our manuscript accompanies this letter. All comments by the reviewer have been addressed. Based on his/her comments, we have made changes to the manuscript, which are detailed below.
|
Reviewer Comment |
Response |
Line # |
|
Reviewer #1 |
||
|
Abstract: Please use past tense |
Abstract has been corrected |
16-25 |
|
Page 2, line 49: provide a reference |
References has been added |
49 |
|
Page 2, line 57/58 “it is necessary … symptoms”: the sentence seems out of context, please delete it or integrate it within the paragraph. |
Sentence has been deleted |
|
|
Page 2, line 85: the aim of “quality of life” is not consistent with the abstract “autonomy and social participation”. |
the aim has been modified consistent with the abstract |
84 |
|
Page 3, line 140: the statement “hearing loss in children common” does not apply to all children. Please specify which children group. |
information is consistent with respect to epidemiology, please see reference |
|
|
Please use past tense across the paper |
verb tenses have been changed |
144-145 148 152-154 |
|
Page 4, lines159-192: This section contains a detailed description of the intervention, but also outcomes of the intervention (e.g., these difficulties disappeared at the end … of treatment”, ...). Although it reads well in its present form, the manuscript would be more organized if this section focused on “intervention” only. I recommend moving the outcomes to the results section. |
|
|
|
The procedure section should clearly state that the outcome measures were used before the intervention, and then after the intervention. |
The specification has been entered |
145-147 |
|
Page 5, line 242: the statement “COTES scores appear to be slightly improved” should read “COTES scores were slightly improved”. |
The change has been made |
242-245 |
|
The patient is sometimes referred to as “he” and sometimes “she”. Please be consistent throughout the paper. |
The change has been made |
262-263 272-273 279-280 |
|
Page 5, starting line 252: the paragraph is hard to follow. The T1 results would be better presented in a table. |
The addition of the table was made |
Table 1 |
|
The entire discussion section needs to be rewritten completely. It lacks the main component of a discussion (main findings of the study, clinical interpretation of results, future research directions, limitations). |
Discussion has been modified |
287-293 |
|
|
|
|
|
Reviewer #2 |
||
|
The manuscript brings interesting content for the current literature regarding occupational therapy approaches in patients with leukodystrophies. The main concern related to the manuscript content is the structure of the text. The discussion of the manuscript should include aspects comparing features observed in the reported case with previously described cases in the literature. Genes mentioned in the text would be presented in italics. Similarly, it is interesting that the authors describe the observed pathogenic variants that defined the diagnosis (I understand this was not the focus of the study, however it can be of importance in future clinical and genetic correlations). |
Discussion has been modified. This study focuses on the importance of rehabilitative intervention, specifically occupational therapy, which is why it would be out of context to specify pathogenic variants that defined the diagnosis. |
287-293 |
We hope that the new version of our manuscript is acceptable for publication.
Best regards,
Donatella Valente

Reviewer 2 Report
The manuscript brings interesting content for the current literature regarding occupational therapy approaches in patients with leukodystrophies. The main concern related to the manuscript content is the structure of the text. The discussion of the manuscript should include aspects comparing features observed in the reported case with previously described cases in the literature. Genes mentioned in the text would be presented in italics. Similarly, it is interesting that the authors describe the observed pathogenic variants that defined the diagnosis (I understand this was not the focus of the study, however it can be of importance in future clinical and genetic correlations).
Author Response
We appreciate the opportunity to resubmit our article entitled “Occupational Therapy intervention in the child with leukodystrophy: case report”. We would like to thank the referees for the careful and constructive reviews. We have made corresponding changes directly to the manuscript where appropriate with changes tracked. The revised version of our manuscript accompanies this letter. All comments by the reviewer have been addressed. Based on his/her comments, we have made changes to the manuscript, which are detailed below.
|
Reviewer Comment |
Response |
Line # |
|
Reviewer #1 |
||
|
Abstract: Please use past tense |
Abstract has been corrected |
16-25 |
|
Page 2, line 49: provide a reference |
References has been added |
49 |
|
Page 2, line 57/58 “it is necessary … symptoms”: the sentence seems out of context, please delete it or integrate it within the paragraph. |
Sentence has been deleted |
|
|
Page 2, line 85: the aim of “quality of life” is not consistent with the abstract “autonomy and social participation”. |
the aim has been modified consistent with the abstract |
84 |
|
Page 3, line 140: the statement “hearing loss in children common” does not apply to all children. Please specify which children group. |
information is consistent with respect to epidemiology, please see reference |
|
|
Please use past tense across the paper |
verb tenses have been changed |
144-145 148 152-154 |
|
Page 4, lines159-192: This section contains a detailed description of the intervention, but also outcomes of the intervention (e.g., these difficulties disappeared at the end … of treatment”, ...). Although it reads well in its present form, the manuscript would be more organized if this section focused on “intervention” only. I recommend moving the outcomes to the results section. |
|
|
|
The procedure section should clearly state that the outcome measures were used before the intervention, and then after the intervention. |
The specification has been entered |
145-147 |
|
Page 5, line 242: the statement “COTES scores appear to be slightly improved” should read “COTES scores were slightly improved”. |
The change has been made |
242-245 |
|
The patient is sometimes referred to as “he” and sometimes “she”. Please be consistent throughout the paper. |
The change has been made |
262-263 272-273 279-280 |
|
Page 5, starting line 252: the paragraph is hard to follow. The T1 results would be better presented in a table. |
The addition of the table was made |
Table 1 |
|
The entire discussion section needs to be rewritten completely. It lacks the main component of a discussion (main findings of the study, clinical interpretation of results, future research directions, limitations). |
Discussion has been modified |
287-293 |
|
|
|
|
|
Reviewer #2 |
||
|
The manuscript brings interesting content for the current literature regarding occupational therapy approaches in patients with leukodystrophies. The main concern related to the manuscript content is the structure of the text. The discussion of the manuscript should include aspects comparing features observed in the reported case with previously described cases in the literature. Genes mentioned in the text would be presented in italics. Similarly, it is interesting that the authors describe the observed pathogenic variants that defined the diagnosis (I understand this was not the focus of the study, however it can be of importance in future clinical and genetic correlations). |
Discussion has been modified. This study focuses on the importance of rehabilitative intervention, specifically occupational therapy, which is why it would be out of context to specify pathogenic variants that defined the diagnosis. |
287-293 |
We hope that the new version of our manuscript is acceptable for publication.
Best regards,
Donatella Valente

Round 2
Reviewer 2 Report
I have no additional comments or suggestions at this point.